# Trend and burden of neural tube defects among cohort of pregnant women in Ethiopia: Where are we in the prevention and what is the way forward?

**Anteneh Berhane**[1,2]*, **Tefera Belachew**[2]

1 Department of Public Health, College of Medicine and Health Science, Dire Dawa University, Dire Dawa, Ethiopia, 2 Department of Nutrition and Dietetics, Faculty of Public Health, Institute of Health Science, Jimma University, Jimma, Ethiopia

* antishaction@gmail.com

**Data Availability Statement:** All relevant data are within the paper and its Supporting Information files.

## Abstract

### Introduction

Neural tube defect is one of the top five most serious birth defects in the world. In Ethiopia an accurate estimate of the trend and burden of neural tube defects is still unknown. There hasn't been much research done on the prevalence and trend of neural tube defects in Eastern Ethiopia. To complement previous efforts of studies, the purpose of this study is to estimate the trend and burden of neural tube defects in Eastern Ethiopia as well as to investigate the epidemiological implications of the findings.

### Methods

A facility-based retrospective cohort study was carried out from cohort pregnant women who delivered in selected hospitals. File records of all babies who were found to have neural tube defects could be reached between 2017 and 2019. A structured checklist was used to collect data. The incidence of each case was calculated by dividing the number of cases per year by the total number of live births in each hospital. To determine the linear trend of neural tube defects over time, linear trend of Extended Mantel-Haenszel chi-square was performed. Data were presented using frequencies and percentages. Data were analyzed using SPSS for windows version 25.

### Results

A total of 48,750 deliveries were recorded during the three years of the study considered for analyses with 522 women having neural tube defect giving an incidence rate of 107.5 per 10,000 live births in the three years. The most common types of neural tube defects found in the area were anencephaly and spina bifida accounting for 48.1% and 22.6%, respectively. The distribution of neural tube defects varied across the study hospitals, with Adama Medical College Hospital having the highest proportion (46.6%). Over half of the mothers (56.7%) live in cities. Mothers in the age group 25–34 (46.9%) and multigravida mothers

**Funding:** This study obtained funds from Dire Dawa University and Jimma University for data collection only. The authors declare that they have no fund for the publication of this manuscript. The funders have no role in the study design, data analysis and decision to publish for preparation of the manuscript.

**Competing interests:** We declare here in that we have no conflict of interest. The funders had no role in the design of the study; in the collection, analyses, or interpretation of data; in the writing of the manuscript, or in the decision to publish the results.

had higher proportions (64.4%).of neural tube defects. None of the mothers took folic acid before conception, and only 19% took iron folic acid supplementation during their pregnancy.

## Conclusion and recommendation

The findings showed that an increasing trend and burden of neural tube defects and preconception folic acid supplementation is insignificant in the region which showed that where we are in the prevention of neural tube defects. The finding suggests that preconception folic acid supplementation in conjunction with health care services should be considered to reduce the risk of neural tube defects in the region. Aside from that, intensive prevention efforts for long-term folate intake through dietary diversification and appropriate public health interventions are required. Furthermore, data must be properly recorded in order to address disparities in neonatal death due to neural tube defects, and the determinants of neural tube defects should be investigated using large scale prospective studies with biomarkers.

## Background

Neural tube defect (NTD) is among the top five most common and serious birth defects of the brain and spinal cord, caused by the failure of the neural tube to close between 21 and 28 days after conception, usually before a woman realizes she is pregnant. The defect ranges from anencephaly through encephalocoeles to spina bifida [1–5]. NTDs are one of significant causes of infant and child mortality, morbidity and long-term disability as well as psychological and great emotional impact on affected families [1]. According to the World Health Organization (WHO), approximately 400 000 births with neural tube defects (NTDs) occur each year, resulting in an estimated 270,000 newborn deaths worldwide [6] causing more than 10% of newborn deaths. Both developing and developed countries bear the burden of NTDs. In countries where folic acid supplementation is not available, the prevalence ranges between 0.5 and 2 per 1000 births. Although the prevalence of NTD varies greatly depending on geography and socioeconomic status [7, 8], it is the leading causes of neonatal deaths in low and middle-income countries, accounting for 29% of all neonatal deaths [9].

In Ethiopia, few studies reported that, the prevalence is increasing from year to year with spatial variations in the increase. The incidence rate ranged from 61/10,000 in Addis Ababa [10] to 131/10,000 in Tigray [11]. The overall burden of neural tube defect in Ethiopia is unknown and underestimated owing to insufficient and fragmented data. Because NTDs are major causes of death among children under the age of five, adequate data are required for well-established interventions. There is currently no evidence on the trend and prevalence of neural tube defects in Eastern Ethiopia. This retrospective analysis provides clues on magnitude and trend of NTD in eastern Ethiopia and it gives insight where is the country in prevention of NTDs also align in the context of intervention efforts on micronutrient prevention and control that government has been exercising since 2005.

## Material and methods

### Study setting

The study was conduct in Dire Dawa City Administration, Harari Regional State and Adama city which are located in the Eastern part of Ethiopia. Dil chora Referral Hospital is found in

Dire dawa located 515 km to the east of Addis Ababa and serves approximately five million populations from Dire Dawa and neighboring Oromia and Somali regions. Hiwot Fana Specialized Teaching Hospital is found in Harar City which is 526 kilometers away from Addis Ababa to the east and delivers services to the entire community of eastern Ethiopia. In addition, the hospitals also serve as teaching centers for health and medical sciences students. Adama Hospital Medical College serves as a referral center for more than 6 million people from different regions and neighboring zones and regions including Afar, Amhara and Somali.

## Study design

A retrospective cohort study was carried out based on reviewing the medical records of a cohort of pregnant women who delivered in Dil-Chora Referral Hospital, Hiwot Fana Specialized teaching Hospital and Adama Medical College Hospital.

## Participant selection

The study hospitals were selected purposefully based on referral status and cases load in the eastern part of Ethiopia. From the total delivered babies in the selected hospitals, all recorded babies delivered, treated, and terminated that diagnosed as having NTDs cases were retrieved from medical admission log-book retrospectively from September 1, 2017 to August 30, 2020 were included. Exclusion criteria included absence of client card, unclear recorded or the client card that had incomplete documentation and had more than 50% of the values missing. The detailed methods of define the target participants were as follows (**Fig 1**).

## Data collection method

A pretested and structured questionnaire developed after relevant literature review was used to retrieve the data. The questionnaire was designed to obtain data that encompasses such as, some demographic, gestational age at the time of birth, use of folic acid and medication during or early pregnancy, hypertension, diabetes and other maternal diseases and time of diagnosis of NTDs. Data were collected from routine administrative hospital records. All NTDs cases were retrospectively reviewed in a sequential manner from admission log books, obstetrics and gynecology wards, and Neonate Intensive Care Unit (NICU). The diagnoses were confirmed by gynecologists, pediatricians, midwives and specialist nurses. Medical Record Numbers (MRN) was used to identify study participants from admission log book. Data were collected via interviewer-administered tablet-based questionnaires using KoBoTool platform.

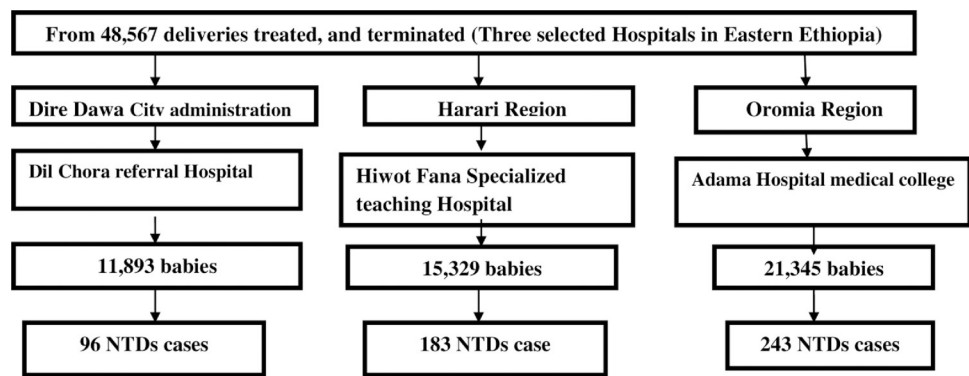

**Fig 1. Schematic presentation of sampling procedure in Eastern 2017–2019.**

Six diploma midwives data collectors and 3 BSc midwives were used to collect data. To ensure data quality a two days training was given on the study's overall procedure to data collectors and supervisors. Permission to access the data was given by the city administration health bureau and hospital administrations.

## Variables

**Dependent variable.** Trend and burden of NTDs.

**Independent variable.** Socio-demographic, pregnancy, ANC use, folic acid and IFA, maternal obstetric history, maternal health and drug history.

## Operational definitions

**NTDs cases.** Is defined as mothers who gave birth to an alive newborn with any type of NTDs (anencephaly, spina bifida, or encephalocele, or myelomeningocele or meningocele), irrespective of gestational age.

**NTDs-affected pregnancy.** Is defined as one of the following four outcomes: (1) an early fetal loss or miscarriage (defined as a spontaneous pregnancy loss at 20 completed weeks of gestation), (2) fetal death or stillbirth (defined as a spontaneous pregnancy loss at 20 completed weeks of gestation), (3) elective termination of pregnancy for fetal anomaly (eTOPFA), or (4) an affected live birth.

**NTDs incidence (burden) was calculated as.**

$$= \frac{affected\ live\ births + affected\ still\ births + eTOPFAs\ for\ NTDs}{Live\ births} \times 10,000$$

**Multiple neural tube defects (MNTDs).** Defined by the simultaneous occurrence of more than one NTD in a single case with "normal" neural tissue in between.

## Data processing and analysis

The data were cleaned and edited before analyses using SPSS for windows version 25. Descriptive statistics was employed to summarize socio-demographic characteristics and estimate the incidence of patients with neural tube defects. The trend of NTD was determined for the years between 2017 and 2019. The burden was calculated by dividing the number of neural tube defect cases identified (numerator) by the total number of births in selected hospitals between 2017 and 2019. Each study site's linear trend was also computed using the corresponding number of live births by year and study site as the denominator. To determine the linear trend of NTDs over time, Extended Mantel-Haenszel chi-square was used.

## Ethical consideration

The study was approved by Jimma University's Institutional Review Board (IRB) with ethical clearance letter number JU/EC/17/0390 as well as waiver of documentation of consent was obtained from the ethics committees of each region and hospitals. Written informed consent was obtained from midwifes and nurses of selected hospitals. No additional patient consent was required. To maintain confidentiality, all information was kept anonymous and adhered to the ethical code for human subjects enshrined in the Helsinki Declaration [12].

## Results

### Socio demographic characteristics

Between 2017 and 2019, a total of 48,567 pregnant women delivered in the three selected hospitals, with 522 neonates having one or more types of NTDs. The overall burden of NTDs was 107.5 per 10,000 live births (live birth and stillbirths, foetal deaths). The distribution of NTDs varied between the hospitals studied such that Adama Medical College Hospital accounted for the highest proportion of cases (46.6%). Over half of the mothers (56.7%) lived in urban areas. Nearly one-third (30.5%) of the mothers lived in East Harerghe, and the mean age of the participants was 26.4 (±5.6 SD), with maternal age 25–34 accounting for 46.9% (Table 1).

### Reproductive and ANC history

Majority (98.9%) of the mothers gave a single neonate, while 64.4% were multigravida. A little more than half of the mothers (51.1%) had ANC follow-up. All mothers did not receive folic acid supplementation throughout the entire pregnancy. Similarly, 81% of mothers did not receive iron and folic acid supplementation throughout their pregnancy. Whereas, only 5.6% of mothers received folic acid contain multivitamin supplement during their pregnancy (Table 2).

### Illness and drug history

The major illnesses identified in the mothers' morbidity history were spontaneous abortion (18.8%), chronic hypertension (1.1%), diabetic mellitus (1.3%), anemia (2.5%), preeclampsia (2.1%), fever (1.3%), viral infection (1.3%), and parasitic infection (0.8%). Furthermore, 2.1% of mothers had a previous history of NTDs, and 0.6% of mothers were living with HIV/AIDS, only 1.1% used an antiepileptic drug (AED) and 2.1% of mothers used antibiotics (Table 3).

**Table 1. Background characteristics and proportion of deliveries with NTDs in the Eastern Ethiopia based on hospital data from 2017–2019.**

| Variables | Frequency | Percent |
|---|---|---|
| **Study hospitals** | | |
| Dil Chora Referral Hospital | 96 | 18.4 |
| Hiwot Fana Specialized Teaching Hospital | 183 | 35.1 |
| Adama Medical College Hospital | 243 | 46.6 |
| **Participant address** | | |
| Dire Dawa | 86 | 16.5 |
| Adama | 149 | 28.5 |
| Eastern Harerghe | 159 | 30.5 |
| Hareri | 21 | 4.0 |
| Somali | 7 | 1.3 |
| West Harerghe | 5 | 1 |
| Other (around Adama) | 95 | 18.2 |
| **Residence** | | |
| Rural | 226 | 43.3 |
| Urban | 296 | 56.7 |
| **Mean maternal age (years)** | 26.4± 5.6 | |
| **Maternal age** | | |
| 18–24 | 205 | 39.3 |
| 25–34 | 245 | 46.9 |
| 35–45 | 72 | 13.8 |

**Table 2. Reproductive and ANC characteristics of pregnant women, Eastern Ethiopia, data from 2017–2019.**

| Variables | Characteristics | Frequency | Percent |
|---|---|---|---|
| Type of pregnancy | Single | 516 | 98.9 |
| | Twins | 6 | 1.1 |
| Gravidity | Primigravidity | 186 | 35.6 |
| | Multigravidity | 336 | 64.4 |
| History of spontaneous abortion | Not documented | 424 | 81.2 |
| | Yes | 98 | 18.8 |
| History of Preterm | Not documented | 521 | 97.9 |
| | Yes | 1 | 0.2 |
| Previous history of NTDs | Not documented | 511 | 97.9 |
| | Yes | 11 | 2.1 |
| Sex affected | Male | 1 | 0.2 |
| | Female | 3 | 0.6 |
| | Not documented | 7 | 1.3 |
| Adverse pregnancy | Not documented | 518 | 99.2 |
| | Yes | 4 | 0.8 |
| Type of adverse pregnancy | APH | 2 | 0.4 |
| | Severe preeclampsia | 2 | 0.4 |
| ANC follow | No | 255 | 48.9 |
| | Yes | 267 | 51.1 |
| Place of ANC Visit | Private clinic/hospital | 80 | 15.3 |
| | Governmental health facility | 180 | 34.5 |
| | Non-governmental health facility | 3 | 0.6 |
| | Not documented | 4 | 0.8 |
| Folic acid supplementation | Not documented/No | 522 | 100 |
| Iron folic acid supplementation | Not documented | 423 | 81 |
| | Yes | 99 | 19 |
| Multivitamin supplementation | Not documented | 464 | 88.9 |
| | Yes | 29 | 5.6 |

APH = Antepartum hemorrhage, ANC = Antenatal care, NTDs = Neural Tube Defects.

## Obstetric history

Extremely preterm (<28 weeks) was the most common gestational age of cases with NTDs. Out of the NTD affected pregnancies, 78.4% were diagnosed by ultrasound before delivery. In terms of mode of delivery, the majority of women had spontaneous vaginal births (87.5%). Nearly equal proportion of males (28%) and females (27.2%) were affected, yielding a sex ratio of 1. Regarding the outcome, 58.2% of NTD-diagnosed pregnancies were terminated medically, while the remaining 27.2% resulted in stillbirths. Only 1.3% of the total newborns with NTDs were discharged alive with referral based on family consent, while the remaining 98.7% died before referral to NICU, delivery, or medical termination (**Table 4**).

## Types of NTDs identified

Anencephaly had the highest proportion (48.1%) of NTDs identified, followed by spinal bifida (22.6%) and myelomeningocele (10.5%) (**Fig 2**).

Nearly a third (27.8%) of the NTD cases were associated with different type of congenital anomalies with most of the congenital anomalies observed in this study being hydrocephalus (79.3%) followed by other type of anomalies (**Fig 3**).

**Table 3. Illness and drug history of pregnant women Eastern Ethiopia data from 2017–2019.**

| Variables | Categories | Frequency | Percent |
|---|---|---|---|
| History of any infection before/early during pregnancy | Not documented | 515 | 98.7 |
| | Yes | 7 | 1.3 |
| Type of infection | Hepatitis B | 2 | 0.4 |
| | Respiratory tract | 1 | 0.2 |
| | UTI | 2 | 0.4 |
| | Urinary tract | 1 | 0.2 |
| | Vulvar edema | 1 | 0.2 |
| Chronic hypertension | Not documented | 516 | 98.9 |
| | Yes | 6 | 1.1 |
| Diabetic mellitus | Not documented | 515 | 98.7 |
| | Yes | 7 | 1.3 |
| History of anemia before/early during pregnancy | Not documented | 509 | 97.5 |
| | Yes | 13 | 2.5 |
| History of preeclampsia | Not documented | 511 | 97.9 |
| | Yes | 11 | 2.2 |
| History of eclampsia | Not documented | 519 | 99.4 |
| | Yes | 3 | 0.6 |
| History of tuberculosis (TB) | Not documented | 521 | 98.8 |
| | Yes | 1 | 0.2 |
| Living with HIV/AIDS | Not documented | 519 | 99.4 |
| | Yes | 3 | 0.6 |
| History of fever | Not documented | 515 | 98.7 |
| | Yes | 7 | 1.3 |
| History of viral infection | Not documented | 515 | 98.7 |
| | Yes | 7 | 1.3 |
| History of parasite infection | Not documented | 518 | 99.2 |
| | Yes | 4 | 0.8 |
| History of gastric | No documented | 518 | 99.2 |
| | Yes | 4 | 0.8 |
| History of taken antibiotic | Not documented | 511 | 97.9 |
| | Yes | 11 | 2.1 |
| Utilized AED | Not documented | 516 | 98.9 |
| | Yes | 6 | 1.1 |

*UTIs = Urinary tract infections*, **AED** = Antiepileptic Drugs, UTI = Upper Tract Infection.

The overall incidence of NTDs was 107.5 per 10,000 live births with the incidence rate showing an increasing trend over a three-year period. The proportion of NTDs increased linearly over three years, with odd ratios (OR) of 1 (2017) and 4.3, and 8.3 for 2018 and 2019, respectively. Extended Mantel-Haenszel chi-square for linear trend is 200.53 (P<0.0001) (**Table 5**).

Hiwot Fana Specialized Teaching Hospital had the highest overall incidence of any of the study hospitals (119.4 per 10,000 births). In 2017 and 2018, Dil Chora Hospital had the highest burden of NTDs cases, with an incidence of 51.3 and 115 cases per 10,000 births, respectively. In 2019 the highest burden of NTDs with an incidence of 244 per 10,000 births was found in Hiwot Fana Specialized Teaching Hospital (**Table 6**).

**Table 4. Obstetric history of pregnant women, Eastern Ethiopia, data from 2017–19.**

| Variables | Categories | Frequency | Percent |
|---|---|---|---|
| Gestational age | Extremely preterm (< 28 weeks) | 254 | 48.7 |
| | Very preterm (28–31 weeks) | 107 | 20.5 |
| | Moderate preterm (32–36 weeks) | 41 | 7.9 |
| | Extremely term (37–38 weeks) | 39 | 735 |
| | Full term (39–40 weeks) | 21 | 4 |
| | Post term (40 weeks) | 1 | 0.2 |
| | Not documented | 59 | 11.3 |
| Mode of NTDs identified | Identified by ultrasound before delivery | 409 | 78.4 |
| | Identified after delivery | 58 | 11.1 |
| | Not documented | 55 | 10.5 |
| Mode of delivery | Spontaneous vaginal | 457 | 87.5 |
| | Cesarean section | 49 | 9.4 |
| | Vacuum | 15 | 2.9 |
| | Forceps | 1 | 0.2 |
| Date of birth | 2017 | 33 | 6.3 |
| | 2018 | 165 | 31.6 |
| | 2019 | 324 | 62 |
| Sex of neonate | Male | 146 | 28 |
| | Female | 142 | 27.2 |
| | Not documented | 234 | 44.8 |
| Birth or pregnant outcome | Stillbirth | 142 | 27.2 |
| | Alive | 62 | 11.9 |
| | Terminated/elective | 304 | 58.2 |
| | Spontaneous abortion | 14 | 2.7 |
| Status of neonate during discharge | Alive | 7 | 1.3 |
| | Dead | 515 | 98.7 |

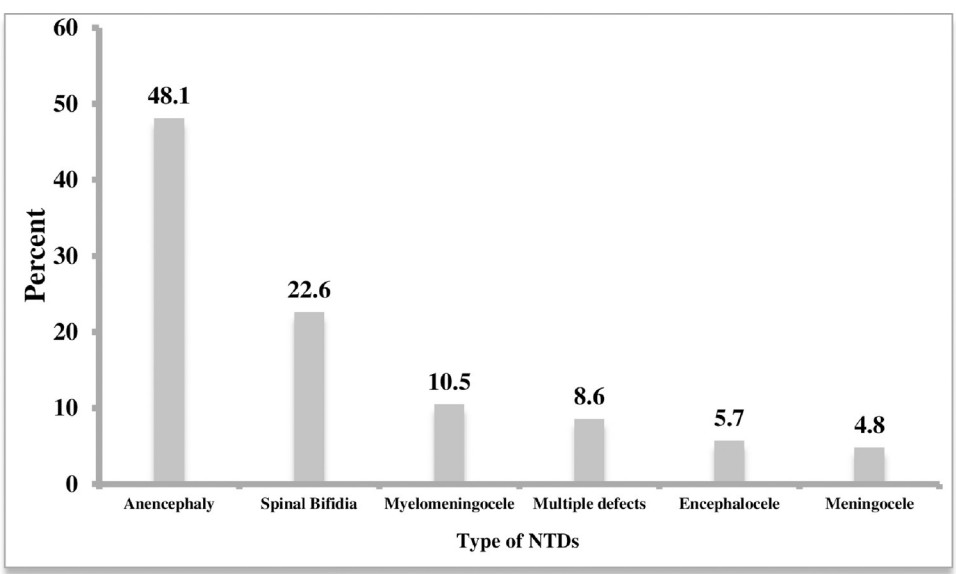

**Fig 2. Type of identified NTDs in Eastern Ethiopia, data from September 2017–2019.**

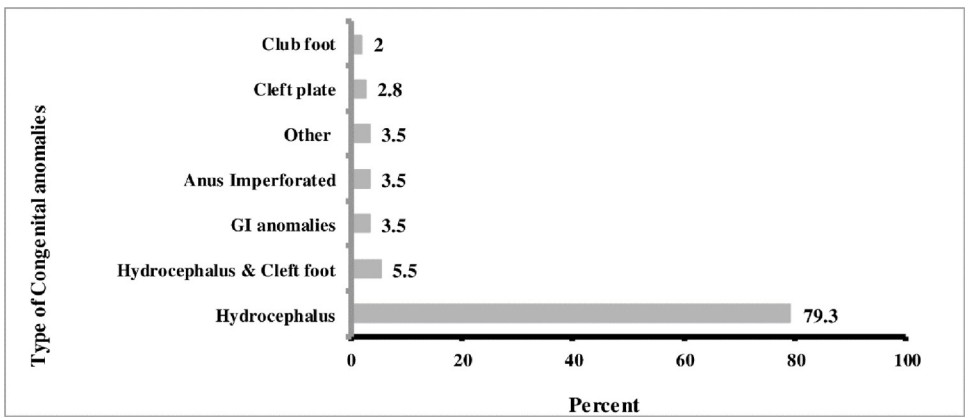

**Fig 3. Type of congenital anomalies associated with NTDs Eastern Ethiopia, data from 2017–2019.**

As depicted in **Table 7**, anencephaly had the highest overall incidence, followed by spina bifida and myelomenigocele, with incidences of 51.7 and 24.3/10,000 births, respectively. Encephalocele and meningocele had the lowest incidences, with 6.2 and 5.2/10,000, respectively (**Table 7**).

**Fig 4** depicts the linear trend of the different types of NTDs over the study period. The occurrence of anencephaly and spina bifida increased steadily, reaching a peak in 2019 (50.3% and 28.4%, respectively), while the occurrence of multiple defects peaked in 2019 (11.5%) (**Fig 4**).

Large proportion of anencephaly (43.4%) cases was found at Hiwot Fana Specialized Teaching Hospital, while the majority of spinal bifida (68.6%) were found in Adama Medical College Hospital. Similarly, Dil-Chora Hospital had the highest proportion of myelomenongocele (49.1%). Hiwot Fana specialization teaching Hospital and Adama medical college Hospital each had 46.7 percent and 43.3% of the total cases of enencephale, respectively. Menengocele was found in higher proportions in Hiwot fana (52%) and Dil chora Hospital (28%) hospitals (**Fig 5**).

East Hararghe had a higher proportion of pregnancies with anencephaly (18.4%) than Adama (11.4%). Regarding spinal Bifdia, 11.5 percent, 4.2%, and 3.4% of mothers were from Adama, around Adama, and Dire Dawa, respectively. Similarly, the majority of myelomenongocele cases were reported in Dire Dawa and East Harerghe (3.8%), while East Harerghe (46.4%) and Adama (30%) had the highest proportion of enencephale cases, East Harerghe (40%) and Dire Dawa (30 percent) had the highest proportion of menengocele cases (24%) (**Fig 6**).

The proportion of mothers who did not receive iron and folate supplementation and had at least one of the NTDs ranged from 68.0 percent to 88.4%. Similarly, the percentage of mothers

**Table 5. Linear trend of NTDs incidence Eastern Ethiopia, data from September 2017–2019.**

| Year | No. of newborns | No. of newborns with NTDs | Proportion | Incidence per 10,000 births | Mantel-Haenszel Summary Odds Ratio |
|---|---|---|---|---|---|
| 2017 | 14479 | 33 | 0.22 | 22.8 | 1 |
| 2018 | 16906 | 165 | 0.97 | 97.6 | 4.3 |
| 2019 | 17182 | 324 | 1.88 | 188.56 | 8.3 |
| Total | 48567 | 522 | 1.07 | 107.5 | |

NTDs = Neural tube defects, extended Mantel-Haenszel chi-square for linear trend is 200.53 (P<0.0001).

**Table 6. Linear trend of NTD incidence among study hospitals Eastern Ethiopia, data from September 2017–2019.**

| Year | Study Hospitals | | | | | | | | |
|---|---|---|---|---|---|---|---|---|---|
| | Adama Medical College Hospital | | | Hiwot Fana Specialization Teaching Hospital | | | Dil Chora Hospital | | |
| | Total delivery | Case | Incidence/10,000 | n | Case | Incidence /10,000 | n | Case | Incidence/10,000 |
| 2017 | 5455 | 1 | 1.8 | 5124 | 12 | 23.4 | 3900 | 20 | 51.3 |
| 2018 | 7584 | 66 | 87 | 5411 | 54 | 99.8 | 3911 | 45 | 115 |
| 2019 | 8306 | 176 | 211.89 | 4794 | 117 | 244.05 | 4082 | 31 | 75.9 |
| Total | 21345 | 243 | 113.8 | 15329 | 183 | 119.4 | 11893 | 96 | 80.7 |

with one or more affected NTDs who had a history of spontaneous abortion prior to the current pregnancy ranged from 10% to 25.6% (**Table 8**).

Both rural and urban mothers had a high burden of anencephaly, accounting for 61.1% and 38.2%, respectively. Anencephaly was the most frequent NTDS in the age groups of 18–24 and 25–34, accounting for 52.7% and 44.5%, respectively. Multigravida mothers had higher rates of anencephaly (6.7%) and spinal bifida (24.1%) (**Fig 7**).

## Discussion

In this study a total of 48,567 deliveries from the selected hospitals were recorded between 2017 and 2019. Our study presented that the overall incidence rate of NTDs was 107.5 per 10,000 live deliveries. Hiwot Fana Specialized Teaching Hospital had the highest burden of NTDs (244 per 10,000 deliveries). The incidence of NTDs observed in our study is lower than that reported in prospective studies of births at three teaching hospitals in Addis Ababa (126 per 10,000 births) [13] and Tigray region (131 per 10,000 births) [11].

The NTDs incidence documented in our study is also higher than the report from a systematic review and meta-analysis conducted in Ethiopia (63.3 cases per 10,000 children) [14], from a three years retrospective study at two teaching hospitals in Addis Ababa with an incidence of 61 cases per 10,000 [10] and from WHO estimation of 22 per 10,000 births in Ethiopia [15], and eight African countries reported by WHO with 11.7 per 10, 000 births [6]. In Ethiopia, the prevalence of folate deficiency is 46.1%. The prevalence of severe folate deficiency in Dire Dawa and Hareri was reported to be 52.9% and 80.7%, respectively [16]. Thus, the high prevalence of folate deficiency could explain the high burden of NTDs in Eastern Ethiopia. The low prevalence of NTDs reported in most developed and many developing countries may

**Table 7. Incidence of type of NTDs among study hospitals Eastern Ethiopia, data from 2017–2019.**

| Type of NTDs | Study Hospitals | | | | | | | |
|---|---|---|---|---|---|---|---|---|
| | Dil Chora Hospital | | Hiwot Fana Specialization Teaching Hospital | | Adama Medical College Hospital | | Total | |
| | n | Incidence per 10,000 | n | Incidence per 10,000 | n | Incidence per 10,000 | n | Incidence per 10,000 |
| Myelomeningocele | 27 | 22.7 | 19 | 12.4 | 9 | 4.2 | 55 | 11.3 |
| Anencephaly | 36 | 30.3 | 109 | 71.1 | 106 | 49.6 | 251 | 51.7 |
| Encephalocele | 3 | 2.5 | 14 | 9.1 | 13 | 6.1 | 30 | 6.2 |
| Meningocele | 7 | 5.9 | 13 | 8.5 | 5 | 2.3 | 25 | 5.2 |
| Spina bifida | 18 | 1.5 | 19 | 12.4 | 81 | 37.9 | 118 | 24.3 |
| Multiple NTDs | 5 | 4.2 | 9 | 5.8 | 29 | 13.6 | 43 | 8.8 |
| Total | 96 | 8.0 | 183 | 119.4 | 243 | 113.8 | 522 | 107.5 |

NTDs = Neural tube defect.

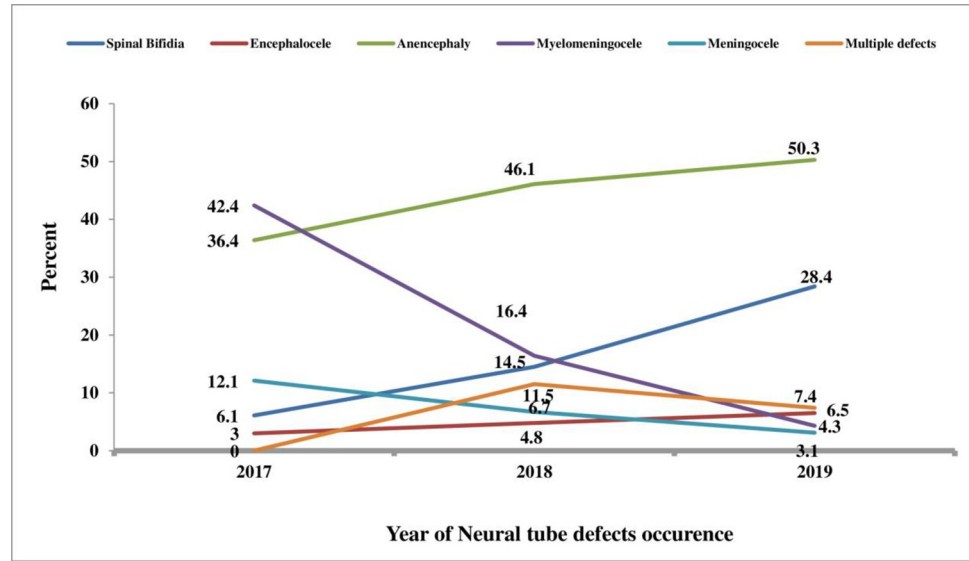

**Fig 4. Yearly distribution of the occurrence of NTDs in Eastern Ethiopia, data from 2017–2019.**

be due to mandatory folic acid fortification [17, 18] and increased health-seeking behavior, health and nutrition adequacy, planned pregnancies, and preconception care services. In contrast, the incidence of 107.5 per 10,000 births observed in our study would be a five-fold increase over the WHO survey estimate in Ethiopia [18]. This alarm indicates the urgent need to implement effective programs to ensure that all women of reproductive age have adequate folic acid on the need to prevent all folic acid-preventable NTDs and the urgent need to implement preconception folic acid supplementation services in Eastern Ethiopia.

Anencephaly was found to be the most common type of NTD (48.1%), followed by spina bifida (22.6 percent), which is consistent with findings from a study conducted at three teaching hospitals in Addis Ababa, Ethiopia [13], in Tigray region, Ethiopia [11], Amhara region, Ethiopia [19], Bale zone Oromia, Ethiopia [20], Gujarat hospital, India (26%) [21], South west Iran (86.8%) [22], in Morocco [23], and in Nigeria [24]. These findings contradict the findings of studies conducted at Tikur Anbessa, Gandhi Memorial, and Ethio-Sewdish Hospitals in

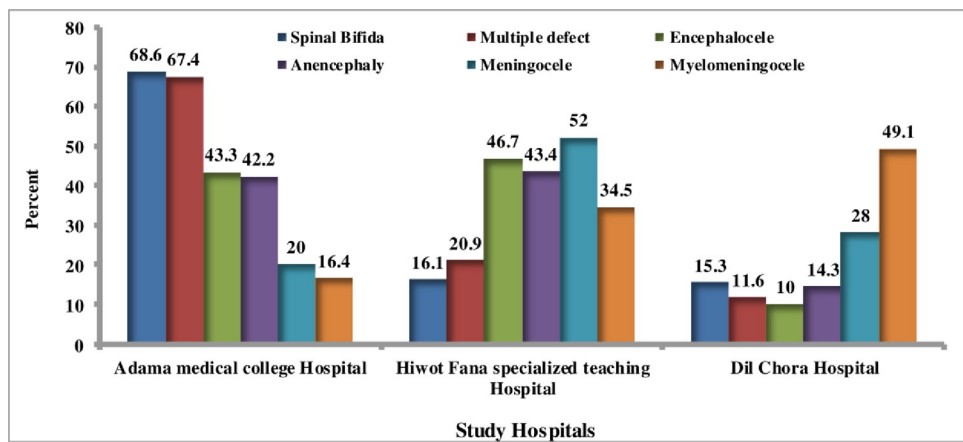

**Fig 5. Percentage of different type of NTDs among study hospital, Eastern Ethiopia, data from 2017–19.**

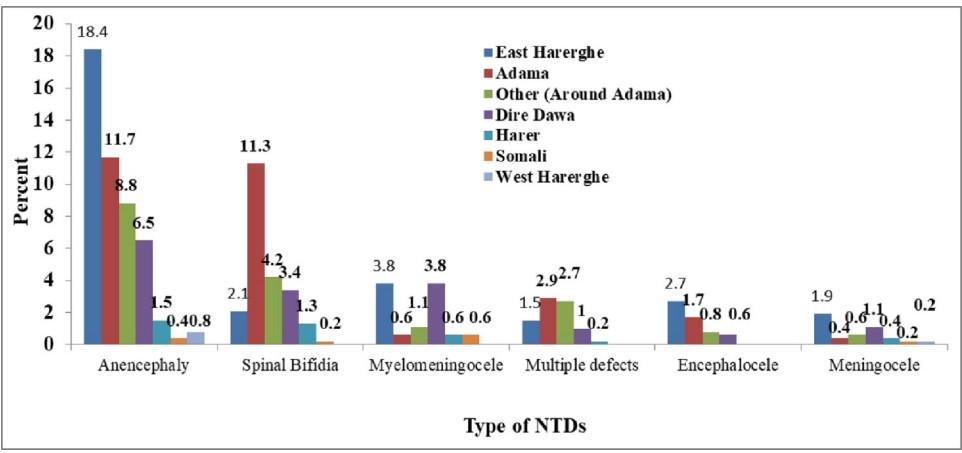

**Fig 6. Distribution of NTDs among study hospitals Eastern Ethiopia, data from 2017–2019.**

Addis Ababa, which reported that the most common NTDs were myelomeningocele and meningocele [10, 25]. This disparity could be attributed to the presence of multifactorial determinants in the various regions and countries where the studies were conducted. In the retrospective studies from the two teaching hospitals in Addis Ababa, Ethiopia, Spina bifida was the most common NTD, followed by anencephaly [10]. The reason why anencephaly is more prevalent than in the previous retrospective study in Addis Ababa is that stillbirths were excluded, whereas our study included stillbirths, and which accounted for 48.7 percent of all NTDs.

In the current study, the distribution of NTDs varied among the study hospitals, with Adama Medical College Hospital accounting for nearly half (46.6%) of cases. This disparity may be due to the fact that more cases around Adama were referred to this hospital due to the presence of different specialist services such as neurologist and the presence of risk factors in the area such as agrochemical exposure. Our study showed that urban resident mothers are more affected than rural residents which accounted for more than half of all NTDs (56.7 percent). This disparity in proportion could be attributed to greater environmental exposure to risk factors in urban areas compared to rural areas, and lifestyle differences between the two setups. This finding contradicts the findings of a study conducted in Amhara Region by Abay W et al., (2020), which revealed that 59.1% and 36.2% of mothers with NTD pregnancy were from rural and urban areas, respectively [19]. Our study found that the sex distribution of

**Table 8. Type of NTDs by FeFol supplementation and history of spontaneous abortion, Eastern Ethiopia data from 2017–19.**

| Type of NTDs | FeFol Supplementation | | History of spontaneous abortion | | Total |
|---|---|---|---|---|---|
| | No | Yes | No | Yes | |
| | n (%) | n (%) | n (%) | n (%) | |
| Myelomeningocele | 38 (69.1) | 17 (30.9) | 49 (89.1) | 6 (10.9) | 55 (10.5) |
| Anencephaly | 203 (80.9) | 48 (19.1) | 202 (80.5) | 49 (19.5) | 251 (48.1) |
| Encephalocele | 25 (83.3) | 5 (16.7) | 27 (90) | 3 (10) | 30 (5.7) |
| Meningocele | 17 (68) | 8 (32) | 21 (84) | 4 (16) | 25 (4.8) |
| Spina Bifida | 102 (86.4) | 16 (13.6) | 93 (78.8) | 25 (21.2) | 118 (22.6) |
| Multiple defects | 38 (88.4) | 5 (11.6) | 32 (74.4) | 11 (25.6) | 43 (8.2) |
| Total | 423 (81) | 99 (19) | 424 (81.2) | 98 (18.8) | 522 (100) |

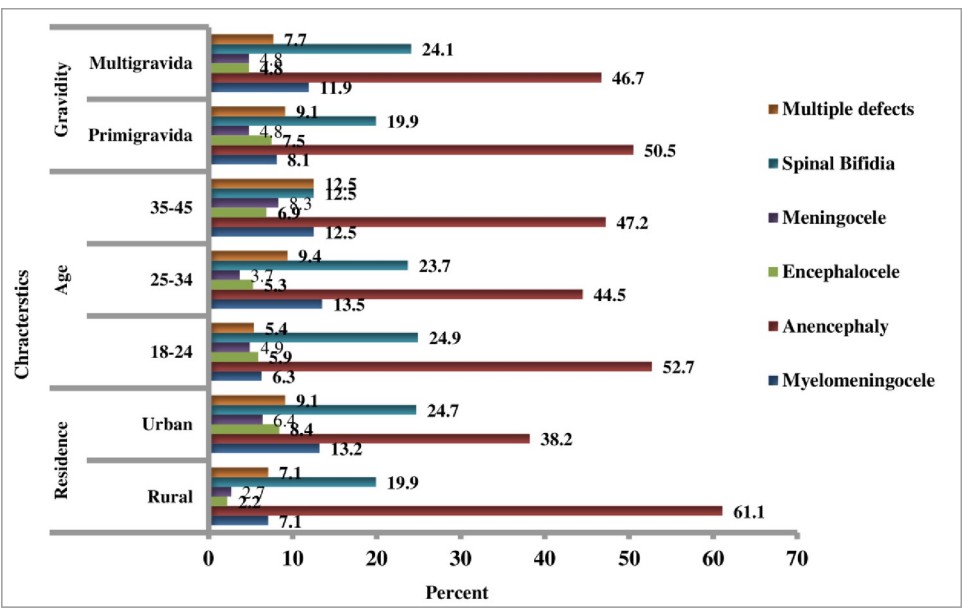

**Fig 7. Type of NTDs by residence, maternal age and gravidity, Eastern Ethiopia, data from 2017–19.**

male and female NTD deliveries was 28 percent and 27.2%, respectively, resulting in a sex ratio of 1:1. Unidentified sex accounted for 44.8%, which is consistent with a study conducted in Thailand, where the sex ratio is 1:1 [26]. This study contradicted the findings of previous studies conducted in Addis Ababa and the Amhara Region of Ethiopia [10, 13, 19] which described female dominance over males. In contrast, a study conducted by Alem et al., (2018) in the Tigray region of Ethiopia found a male predominance over females [11]. There is no single reason why neural tube defects affect more females than males or vice versa.

Our findings also revealed that 18.8 percent of mothers had previously had an abortion. This could be due to trophoblastic cell rest from an earlier aborted pregnancy. This finding is nearly identical (17.3%) to the findings reported by Marco et al., (2011) [27] and Atlaw et al., (2019) at Bale zone Hospitals, South Eastern Ethiopia, which is accounted 47.6% [20].

Preconception folic acid supplementation was found to be protective against NTDs in studies [28–31]. Our research also found that all mothers did not received folic acid supplementation throughout their pregnancy. This finding is consistent with studies conducted in Addis Ababa, Ethiopia, and Morocco [10, 13, 23]. This could be due to is lack of preconception care in the country as well as a lack of media coverage on promotion of preconception of folic acid supplementation. This finding has far-reaching practical implications. After 16 years of implementing micronutrient prevention and control guideline in Ethiopia, such a high incidence of NTD above the WHO cut-off (6/10,000 live births) [32] combined with no supplementation given to all cohorts of pregnant women with NTDs even during pregnancy calls for urgent action. Because NTD occurs at the 28[th] day of pregnancy, strengthening preconception to supplementation of folic acid through various strata should be targeted and researched further.

The following limitations are acknowledged in this study. Because the study was conducted in only three hospitals, it does not represent the true prevalence of NTDs in the community. Determinants of NTDs have not been investigated or attempted. Because this is a retrospective study, there are significant limitations to the recorded data. In some cases, the necessary investigation and complete history were not properly documented. On the other hand, there was a discrepancy between the medical recorded number (log book) and the actual client card,

resulting in the study failing to capture nearly half of the data recorded book in study hospitals. As a result, this study did not provide an accurate magnitude and figure in the study area. Furthermore, because the study focused in the eastern part of the country's the findings may not accurately reflect the national situation and should be interpreted with caution.

## Conclusion

NTD is a significant public health burden in the study area with the most common forms being anencephaly and spinal bifida. The incidence rate is five-fold higher than the WHO estimates for Ethiopia. Moreover, preconception folic acid supplementation is negligible among the study participants and nearly all neonates with NTDs cases were died. The findings suggest the need for strength of primary preventative strategies with active promotion of preconception care service and possible implementation of preconception folic acid supplementation approaches and food fortification with promote having good dietary practice in order to reduce the burden of NTDs as public health emergency in Ethiopia. This will enable the achievement of Sustainable Development Goal 3.2 which states 'end preventable deaths and disabilities in neonates and children under 5 by 2030'. Further investigation of dietary practice of mother who delivered neonate with NTDs or terminated due to NTDs affected pregnancy and the determinants factors of NTDs in the study area with supporting biomarkers is recommended.

## Supporting information

**S1 Checklist. Retrospective data collection tool.**
(DOC)

## Acknowledgments

The authors would like to thank Dire Dawa University and Jimma University, all of the study participants, data collectors, and supervisors who participated in the study, as well as the kind and cooperative staff of the health facilities in eastern Ethiopia.

## Author Contributions

**Conceptualization:** Anteneh Berhane.

**Data curation:** Anteneh Berhane, Tefera Belachew.

**Formal analysis:** Anteneh Berhane, Tefera Belachew.

**Investigation:** Anteneh Berhane.

**Methodology:** Anteneh Berhane, Tefera Belachew.

**Software:** Anteneh Berhane.

**Supervision:** Tefera Belachew.

**Validation:** Anteneh Berhane.

**Writing – original draft:** Anteneh Berhane.

**Writing – review & editing:** Anteneh Berhane, Tefera Belachew.

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
