## [Decision Letter · Decision Letter 0]

19 Nov 2021

PONE-D-21-19307Trend and Burden of Neural Tube Defects among cohort of pregnant women: Where are we in the prevention and what is the way forward?PLOS ONE

Dear Dr. Yaye,

Thank you for submitting your manuscript to PLOS ONE. After careful consideration, we feel that it has merit but does not fully meet PLOS ONE’s publication criteria as it currently stands. Therefore, we invite you to submit a revised version of the manuscript that addresses the points raised during the review process.

We look forward to receiving your revised manuscript.

Kind regards,

Wubet Alebachew Bayih, M.Sc.

Academic Editor

PLOS ONE

Journal Requirements:

2. Please modify the title to ensure that it is meeting PLOS’ guidelines (https://journals.plos.org/plosone/s/submission-guidelines#loc-title). In particular, the title should be "specific, descriptive, concise, and comprehensible to readers outside the field" and in this case please ensure that it is informative and specific about your study's scope and methodology.

*PLOS ONE does not copy edit accepted manuscripts (https://journals.plos.org/plosone/s/criteria-for-publication#loc-5). To that effect, please ensure that your submission is free of typos and grammatical errors.

“The authors acknowledge all study participants, data collectors, and supervisors who took part in the study, as well as the kind and cooperative staff of the health facilities in eastern Ethiopia. Jimma and Dire Dawa University s deserve a special appreciation for the financial support.**”**

“This work was supported by Jimma University and Dire Dawa University. the funders had no role in study design, data collection and analysis, decision to publish or preparation of the manuscript.”

6. Please include a separate caption for each figure in your manuscript.

7. We note you have included a table to which you do not refer in the text of your manuscript. Please ensure that you refer to Table 1, 3 & 5 in your text; if accepted, production will need this reference to link the reader to the Table.

8. Please include a copy of Table 7 which you refer to in your text on page 9.

Additional Editor Comments (if provided):

General comments

Dear authors on your scholarly work; you have brought an important study problem with good findings that have public health importance in the area of practice. However, the manuscript has multiple language usage flaws including punctuations, wordings, spelling and mainly grammar errors. These problems are found throughout the manuscript. Moreover, there are several methodological gaps. Therefore, please make repeated proof-reading and thorough copyediting before considering the manuscript for publication. This would help increase the readability of the manuscript if published.

Specific comments

1.Title: the study area should be included in the title

Abstract

2.Background of the abstract doesn’t clearly show the existing burden of NTD in Ethiopia, and even elsewhere in the globe. Generally, burden of NTD should be numerically stated followed by the objectives showing the research gap the authors would like to address.

3.Methods of abstract should include sampling technique, measurement of NTD, type of data collection tool (adapted or adopted) and software for data entry and analysis.

Background

4.In the last paragraph, it is better to include national incidence of NTD during the launch of different interventions in 2005 in Ethiopia

Methods

5.Add a separate subsection of study area. Then, important details including nutrition culture and ANC follow up of the study population should be clearly stated so that any reader new to the Eastern society can get some understanding of the study area.

6.Why the authors considered only 2017-2019 time period which is actually not sufficient to show the time trend of NTD.

7.Variables and operational definitions: Kindly include a separate subsection detailing measurement of the variables considered for the study.

8.Sampling technique: It would be more self explanatory and easily understandable if the authors showed pictorial presentation (flow chart) of the sampling procedure including how many regions �districts � hospitals � sample size (A cohort of 48,567 pregnant women delivered in three selected hospitals from 2017 to 2019)

9.Please upload your data collection tools (for both quantitative and qualitative) as additional file.

10.Ethical clearance: What beneficent actions will the authors provide the community in return for this study?

Results

11.Please include a separate section that addresses incidence of NTD than mixing it with socio-demographic characteristics section.

Discussion

17.The authors present severity of iron folate deficiency for the high burden of NTD in the study area than other regions. Why folate deficiency in Eastern Ethiopia is severe than other regions? Kindly give strong evidence, because it is clear that iron folate is uniformly distributed to all regions of the country.

Reviewers' comments:

Reviewer's Responses to Questions

**Comments to the Author**

1. Is the manuscript technically sound, and do the data support the conclusions?

Reviewer #1: Partly

Reviewer #2: Yes

2. Has the statistical analysis been performed appropriately and rigorously? 

Reviewer #1: No

Reviewer #2: Yes

3. Have the authors made all data underlying the findings in their manuscript fully available?

Reviewer #1: No

Reviewer #2: Yes

4. Is the manuscript presented in an intelligible fashion and written in standard English?

Reviewer #1: Yes

Reviewer #2: Yes

5. Review Comments to the Author

Reviewer #1: I have some questions for clarification and suggestion for betterment of this article.

1. In your method section, it is not clear that from how many hospitals in eastern Ethiopia, you was selected those three hospitals?

2. I am not sure that your study design clearly retrospective cohort. What makes different from chart/record review?

3. Sample size determination and sampling procedure is not clear.

4. You should clearly describe morbidity/ major illnesses for current pregnancy and previous pregnancy separately.

5. It will be better if you add risk factors for NTDs.

Reviewer #2: Comments to the author

Methods

The study is very relevant and well structured. Just including following few suggestions might be useful.

1.Better to say a facility based instead of” institution based”

2.Why you want to focus from 2017 to 2019? What is new within this period?

3.Are there only 3 hospitals in Eastern Ethiopia?

4.Your study was” among cohort pregnant women who delivered in Dil-Chora, Hiwot Fana specialization teaching Hospital, and Adama Medical College Hospital” but the study population was medical records of cases who delivered or terminated or stillbirth or dead neonate with neural tube defects. Who were your exposed and unexposed groups? Please clarify this statement.

5.“On the other hand, any type of NTDs case which is not clearly recorded and inconsistent data or data with more than 50% of values missing was excluded from the study” did the author excluded inconsistent data within the study period ? If yes how much?

6.Was it closed cohort or open cohort study?

7.The author is required to clarify sample size calculation

8.In your sampling technique” all neural tube defects case those born in selected hospitals were included and selected conveniently” so what is the importance of talking about sample size calculation?

9.Who were your data collectors and supervisors?

6. PLOS authors have the option to publish the peer review history of their article (what does this mean?). If published, this will include your full peer review and any attached files.

Reviewer #1: No

Reviewer #2: No

---

## [Author Response · Author response to Decision Letter 0]

14 Jan 2022

Reviewer #1: 

1. In your method section, it is not clear that from how many hospitals in eastern Ethiopia, you was selected those three hospitals?

Answer

Thank you for your comment. There are more than 20 hospitals in the area but we only focused on the rank of hospital (tier of hospitals based on their service) that means referral, teaching hospitals and caseload because only this hospitals are only given the service regarding to NTDs case. Base on this there are 4 hospitals in that level and we took 3 hospitals. The reason why we left 1 hospital is due to a new upgraded and started after 2017. As we mention, our study started from 2017.

2. I am not sure that your study design clearly retrospective cohort. What makes different from chart/record review?

Answer

As you know the main difference between the chart review and retrospective cohort is that chart review establish whether necessary information is available in the charts and inappropriate for study question. Retrospective studies may be based on chart reviews (data collection from the medical records of patients) and retrospective cohort studies (current or historical cohorts). So based on this facts we used the pretest structured questionnaire for data extraction from the mother and baby client card (including history of mothers) as well medical record book (about history of babies during delivery). 

3. Sample size determination and sampling procedure is not clear.

Answer

Thank you for your question and corrected in the manuscript.

4. You should clearly describe morbidity/ major illnesses for current pregnancy and previous pregnancy separately.

Answer

Thank for your comment but as you know it is a secondary data and we did not found whether the illness was occurred previous pregnancy or current pregnancy. That is why we use the term “History of…”

5. It will be better if you add risk factors for NTDs.

Answer

The risk factor is the next research area and as you read in this manuscript we suggested that it is better to investigate the determinant of developing NTDs in the area. 

Reviewer #2: 

1. Better to say a facility based instead of” institution based”

Answer

Thank you for your comment and corrected in manuscript.

2. Why you want to focus from 2017 to 2019? What is new within this period?

Answer 

As we mentioned in the limitation part we did not found the document before 2017. Even if from 2017 there are significant limitations to the recorded data. In some cases, the necessary investigation and complete history were not properly documented. Already we mention as a limitation of this study and recommended that it should be put the recorded data properly unless the exact figure of the incidence/burden not known which means we don’t know where we on the prevention of NTDs and we don’t have a data that NTDs contribution on neonate mortality.

3. Are there only 3 hospitals in Eastern Ethiopia?

Answer

As we mentioned in the manuscript we select the hospitals based on case load and referral hospital. So based on this criteria we put there are 4 hospitals and we take 3 hospitals. The reason why we left 1 hospital is due to a new upgraded and started after 2017. As we mention, our study started from 2017.

4. Your study was” among cohort pregnant women who delivered in Dil-Chora, Hiwot Fana specialization teaching Hospital, and Adama Medical College Hospital” but the study population was medical records of cases who delivered or terminated or stillbirth or dead neonate with neural tube defects. Who were your exposed and unexposed groups? Please clarify this statement.

Answer

Yes, as you see in the document, data was retrieved from cohort pregnant women, who delivered or terminated with NTD affected pregnancy and history of women was also taken and analysed. We didn’t have a control group, so we are not classified as exposed and unexposed group. That is why we analysed the trend and burden of cases.

5. “On the other hand, any type of NTDs case which is not clearly recorded and inconsistent data or data with more than 50% of values missing was excluded from the study” did the author excluded inconsistent data within the study period ? If yes how much?

Answer

Yes, 14 cases were excluded from the analysis. 

6. Was it closed cohort or open cohort study?

Answer

We believe that our study is closed cohort, because in closed cohort studies risk estimates are assessed in a short time interval as the ratio of cases over those at risk at the beginning of the study (incident proportion or risk in a given interval) and the cohort remains static during the study. So we assess the incidence/burden of NTDs in the study area. 

7. The author is required to clarify sample size calculation

Answer

Thank you for your comment and corrected in manuscript.

8. In your sampling technique” all neural tube defects case those born in selected hospitals were included and selected conveniently” so what is the importance of talking about sample size calculation?

Answer

Thank you for your comment and corrected in manuscript.

9. Who were your data collectors and supervisors?

Answer

Thank you for your comment and included in manuscript.

---

## [Decision Letter · Decision Letter 1]

2 Feb 2022

Trend and burden of neural tube defects among cohort of pregnant women in Ethiopia: Where are we in the prevention and what is the way forward?

PONE-D-21-19307R1

Dear Dr. Berhane,

We’re pleased to inform you that your manuscript has been judged scientifically suitable for publication and will be formally accepted for publication once it meets all outstanding technical requirements.

Kind regards,

Wubet Alebachew Bayih, M.Sc.

Academic Editor

PLOS ONE

Additional Editor Comments (optional):

The authors shall go through their entire revised manuscript for its readiness of publication.

Reviewers' comments:

Reviewer's Responses to Questions

**Comments to the Author**

1. If the authors have adequately addressed your comments raised in a previous round of review and you feel that this manuscript is now acceptable for publication, you may indicate that here to bypass the “Comments to the Author” section, enter your conflict of interest statement in the “Confidential to Editor” section, and submit your "Accept" recommendation.

Reviewer #1: All comments have been addressed

Reviewer #2: All comments have been addressed

2. Is the manuscript technically sound, and do the data support the conclusions?

Reviewer #1: Yes

Reviewer #2: Yes

3. Has the statistical analysis been performed appropriately and rigorously? 

Reviewer #1: Yes

Reviewer #2: Yes

4. Have the authors made all data underlying the findings in their manuscript fully available?

Reviewer #1: Yes

Reviewer #2: Yes

5. Is the manuscript presented in an intelligible fashion and written in standard English?

Reviewer #1: Yes

Reviewer #2: Yes

6. Review Comments to the Author

Reviewer #1: (No Response)

Reviewer #2: (No Response)

7. PLOS authors have the option to publish the peer review history of their article (what does this mean?). If published, this will include your full peer review and any attached files.

Reviewer #1: No

Reviewer #2: No